# Does Generic Cyclic Kinase Insert Domain of Receptor Tyrosine Kinase KIT Clone Its Native Homologue?

**DOI:** 10.3390/ijms232112898

**Published:** 2022-10-25

**Authors:** Julie Ledoux, Luba Tchertanov

**Affiliations:** Centre Borelli, ENS Paris-Saclay, CNRS, Université Paris-Saclay, 4 Avenue des Sciences, F-91190 Gif-sur-Yvette, France

**Keywords:** receptor tyrosine kinase, RTK, KIT cytoplasmic region, kinase insert domain, KID, generic cyclic KID, molecular dynamics and folding, conformational plasticity, intrinsically disordered region, IDR, transient states, free energy landscape

## Abstract

Receptor tyrosine kinases (RTKs) are modular membrane proteins possessing both well-folded and disordered domains acting together in ligand-induced activation and regulation of post-transduction processes that tightly couple extracellular and cytoplasmic events. They ensure the fine-turning control of signal transmission by signal transduction. Deregulation of RTK KIT, including overexpression and gain of function mutations, has been detected in several human cancers. In this paper, we analysed by in silico techniques the Kinase Insert Domain (KID), a key platform of KIT transduction processes, as a generic macrocycle (KID^GC^), a cleaved isolated polypeptide (KID^C^), and a natively fused TKD domain (KID^D^). We assumed that these KID species have similar structural and dynamic characteristics indicating the intrinsically disordered nature of this domain. This finding means that both polypeptides, cyclic KID^GC^ and linear KID^C^, are valid models of KID integrated into the RTK KIT and will be helpful for further computational and empirical studies of post-transduction KIT events.

## 1. Introduction

Most proteins of the human proteome are assembled from several structural (or functional) units. These so-called protein domains form a modular architecture of a macromolecule [1,2]. It is generally accepted that protein structural domains can fold, function, select and be selected independently of the rest of the protein [3]. In many cases, protein domains in isolation can be successfully expressed and tend to fold spontaneously into their native 3D structure while retaining their overall functions. Due to the complexity of using the standard empirical and/or computational biophysical methods for studying large proteins, their splitting on isolated domains is a very attractive strategy. This app roach provides a promising route for using cleaved domains as independent units/blocks in research and biotechnology of large multidomain proteins (MDPs), particularly for studying protein–protein interactions [4]. In these cases, general rules are then derived by looking at the individual domains in detail, assuming that large proteins behave like beads on a string, i.e., the function of a large protein can be understood by summarising the functions of its domains.

In many cases, modular proteins have a hybrid composition. They include structurally well-organized domains connected by intrinsically disordered regions (IDRs), which appear to be crucial elements in the cooperative folding of MDPs and the modulation of protein functions [5,6,7]. As the inherent flexibility of IDRs delivers dynamic cross-domain communication between remote domains, enabling cooperative allosteric regulation in an MDP, the use of cleaved IDRs as stand-alone entities is still questioned [8,9].

Intrinsically disordered proteins (IDPs), including one or several IDRs, are involved in regulatory pathways and cell signalling and sample an extensive range of conformations [6,10,11,12]. Investigation of structural ensembles of IDPs is difficult for both experiment and simulation. On the other hand, if an IDP has a modular architecture in its structures, this property yields a more efficient functional activity performance. It was recognised that modules consist of groups of highly cooperative residues [2,13], which may possess certain functional independence. Usually, protein modules are interconnected through amino acids that maintain the shortest pathways between all amino acids and are, thus, crucial for signal transmission, leading to robust and efficient communication networks [4,8,14,15,16]. This modular organisation is advantageous and, as such, has been conserved in its improved version. Many kinds of structural disorganisations can lead to deteriorated processes prompting dysfunction of normal physiological functions and causing severe diseases [17,18].

Studying the physiologically or pathologically related processes, in particular post-transduction effects, is frequently limited or impossible due to the poor solubility and stability of the associated proteins. From a practical point of view, considering the large size of such proteins and the technological and methodological problems of studying IDPs, certain specific proteins can be analysed per domain.

Receptor tyrosine kinases (RTKs) are the archetypical modular membrane proteins possessing both well-folded and disordered domains acting together in ligand-induced activation and regulation of a post-transduction process that tightly couples extracellular and cytoplasmic events. They ensure the fine-turning control signal transmission from the outside of the cell inward through the cell to the genes by signal transduction [19,20,21].

Deregulation of RTK KIT, including overexpression and gain of function mutations, has been detected in several human cancers. The mutation-induced disorder is directly linked to leukaemia [22,23], in almost all cases of systemic mastocytosis [24] and other hematopoietic cancers; gastrointestinal stromal tumour (GIST) [25], melanoma [26], and others [27].

Similar to all RTKs, KIT contains a tyrosine kinase domain (TKD) crowned by several IDRs–juxtamembrane region (JMR), kinase insert domain (KID), and C-tail [28], which are inherently coupled [29]. In turn, the TKD of KIT is also composed of two sub-domains–N- and C-lobes—enriched by an IDR called activation (A-) loop, tightly collaborating in the activation/deactivation process [30]. Therefore, using these IDRs as independent isolated units instead of their natively fused to TKD states requires careful consideration and further investigation [31,32].

Each KIT IDR contains functional phosphotyrosine residues that act as critical regulatory elements that contribute to KIT activation and/or mediating protein–protein interactions. JMR is the bi-functional segment playing a regulatory role in the activation/deactivation process and the recruitment of signalling proteins. At the same time, KID only participates in the selective recognition and binding of adaptors, signalling and scaffolding proteins [20,21,28]. Multiple functional phosphorylation sites of KID from KIT, three tyrosine (Y703, Y721, Y730), and two serine (S741 and S746) provide alternative binding sites for the intracellular proteins [33]. Phosphorylation of Y703 supplies the binding site for the SH2 domain of Grb2, an adaptor protein initiating the Ras/MAP kinase signalling pathway. Phosphorylated Y721 and Y730 are PI3K and phospholipase C (PLCγ) recognition sites, respectively. The function of Y747 has not yet been described. Phosphorylated serine residues, S741 and S746, bind PKC (protein kinase C) and contribute to the negative feedback of PKC activity under receptor stimulation.

The phosphorylation and binding of KIT domains having multiple functional phosphorylation sites is a great challenge [34,35]. Given its many phosphotyrosines, nothing is known about how such processes occur. Is single-site tyrosine phosphorylation sufficient for a protein to bind (a one-to-one process) to such a domain? Or is protein binding a more collective event, described as multithreaded processes—one to many, or many to one, or many to many—in which, for example, protein binding induces conditions for the phosphorylation of another tyrosine followed by binding of another protein, or partner binding requires phosphorylation of two or more tyrosine sites at the target? To answer these questions, it is necessary to consider many cases of phosphorylation/binding events described by a factorial function. Only for KID with three functional tyrosine residues, the number of combinations analysed is seven. However, if we take into account that in RTK KIT, eight tyrosine phosphorylation sites have been identified in vivo (Y568 and Y570 in JMR; Y703, Y721, and Y730 in KID; Y823 in A-loop; Y900 in the C-lobe; and Y936 in C-tail) [36], as well as two additional sites having been detected in vitro in the activated kinase domain (Y547 and Y553 in JMR) [37], the number of combinations is drastically increased. The modularity of KIT yields a more efficient performance of the functional activity study.

To begin such exploration, even by in silico methods, a single modular domain should be carefully determined and optimised before studying phosphorylation effects. Our recent in silico study (3D de novo modelling and molecular dynamics (MD) simulations) suggested that the cleaved KID (isolated protein) better reproduces the natively fused KID if simulated with locally N- and C-ends to mimic the native steric condition [38]. To deliver the usable species as an initial template for the empirical studies, a generic cyclic KID of RTK KIT, composed of the 80-amino-acid cleaved polypeptide (F689–D768) cyclised by insertion of four Gly residues acting as a physical connector or spacer between its N- and C-KID termini, was proposed as an entity that would be best suited for future studies on the KIT post-transduction effects involving KID. We suggested that this generic cyclic KID (KID^GC^) is also an intrinsically disordered protein (IDP). As the characterisation of IDPs is not a trivial problem, in this paper, we report the structural description of the KID and resort to the recapitulation of the available KIT KID data obtained for the different KID species to compare them in terms of Gibbs free energy.

The characterisation in terms of structural and biophysically related metrics of the conformational spaces generated by the large-scale MD simulations of KID which was considered to be (I) a generic macrocycle (KID^GC^), (II) a cleaved isolated polypeptide (KID^C^), and (III) a natively TKD-fused domain (KID^D^) [29,39] (Figure 1), inspired us to examine the crucial question: how does a KID evolve when studied in isolation compared to more complex architectures? While the allostery of a multidomain protein and the role of quaternary structure in modulating affinity is well established in many proteins [40], we asked whether the folding of the KID^GC^ and its binding sites are in correct position for its functioning.

## 2. Results

A 3D model of the KID generic macrocycle (KID^GC^) was obtained from the randomly chosen MD conformation of the KID polypeptide from a restrained isolated KID. Composed of 80 amino acids (F689–D768) by integration of a short spacer constituted of four glycine residues (GGGG motif). An optimised and well-equilibrated model of KID^GC^ was studied by extended (2-µs) MD simulation running twice in strictly identical conditions. As a newly produced species, KID^GC^ was first characterised in terms of conventional descriptors using generated MD replicas and further compared with native KID’s fused to TKD of KIT (KID^D^) and cleaved KID simulated as an isolated polypeptide with easy restrained end-to-end distance (induced pseudo-rigid constraints) (KID^C^).

### 2.1. General Characterisation of KID^GC^

The root-mean-square deviations (RMSDs) calculated on KID^GC^ conformations from two replicas in the same initial structure show good convergence (Figure 2A). Compared with the great and fast variations of RMSDs in KID^D^ and KID^C^, associated with the significant conformational transitions [38], the RMSD curves of KID^GC^ are significantly smoothers and vary within 2–3 Å.

Similar to the other KIDs, according to DSSP [41], KID^GC^ shows an essential portion of the helical fold (45–51%), which is composed of 5–6 transient helices frequently transforming into the other structural motifs (α-helix↔3_10_-helix↔turn/bend) (Figure 2C,D). A clear predominance of α-helix, two times more frequent than 3_10_-helix, is evident. Nevertheless, the overall occurrence of each helix computed on the concatenated trajectories is dissimilar: 80% for H1 and H6 and only 50–70% for other helices—H2–H5 (Figure 2E). Comparing the KIDs’ helicity, we noted that the GGGG spacer significantly increased the helical content in KID^GC^ concerning KID^D^ and KID^C^ having the portion of the helically folded residues of 25–30 and 30–35%, respectively [38].

The elastic GGGG motif retains the dynamical ability of the N- and C-terminals residues of KID^GC^ characterised by an inter-distance from 5 to 15 Å, likely in KID^D^ and KID^C^ [38], and mean value (m.v.) of 10 Å, corresponding precisely to the value observed in all KIT crystallographic structures [39,42]. The root-mean-square fluctuation (RMSFs) curves subdivide the KID^GC^ residues into two groups, characterised by high and small RMDF values. Additionally, the groups of residues showed the extreme amplitude of fluctuations, either the highest or lowest, which are conserved in different KID. We found early on that the weakly fluctuating residues are involved in the multiple non-covalent interactions stabilising the globule-like shape of KID [38]. Here, we focus on the KID’s highly fluctuating residues. We suggest that these residues may be the main factors that influenced the conformational diversity of KID. In addition to the highly fluctuating N- and C-terminal residues interconnected in KID^GC^ by the elastic spacer GGGG, three other segments, C714-M722 (1), R739-V742 (2) and E758 (3), systematically show the enlarged RMSF values during MD simulations (Figure 2E). These residues are either from the unregular (random coil) or partially folded transient structures.

To estimate the residues fluctuations concerning a stable αH1-helix, taken as reference structure, and visualise the variance, the position of each mid-point residue (Cα-atom) of the maximally fluctuating KID^GC^ residue was aligned on αH1-helix (A701-N705) and projected into KID 3D structure. First, the maximally fluctuating residues are nearly equidistant from the KID^GC^ H1-helix (RMSF value of ≈ 6.5 Å), and their spatial position is described as the elongated surface distributions with an apparent shape of the oblate spherical sector comparable for all maximal fluctuation residues both in length and area occupied differed in spatial position.

Most instances of C691 are distributed along the y–z plane. R740, E758, and E767 are distributed mainly on the x–y plane (Figure 2F). As was expected, distributions formed by N- and C-terminals (C691 and E767) linked by the GGGG motif are closely positioned. Distributions of the highly fluctuating residues from segments 1, 2, and 3 are mutually orthogonal and, together with the N- and C-terminal arrays, represent a spiral galaxy form, as viewed at the top.

The tyrosines—key KID residues—show highly variable spatial positions. Distances connecting the apexes of a tetrahedron designed on the Cα-atoms of four tyrosine residues display very fluctuating values (Figure 3A,B). Distances Y730-Y747, Y703-Y730, Y721-Y730, and Y703-Y721 represent the asymmetric bimodal skew-normal distributions of quasi-equivalent probability, with a minimal contribution (<0.1) of the second components.

The main features of these distributions differed only in their maxima values position—at 8, 11, 14 and 16 Å, respectively. Two other tetrahedron distances, Y721-Y747 and Y703-Y747, are described as multimodal distributions—bimodal (Y721-Y747) with the maxima at 8 and 15 Å, and three-modal (Y703-Y747) with the utmost at 11, 13, and 17 Å.

The tyrosine residues (Cα-atoms) projected into KID 3D structure after the alignment on Y703 at αH1 helix display different spatial distributions—the compact for Y730 (red), more enlarged for Y747 (green), and broad and subdivided on the separated clusters for Y721 (lilac) (Figure 3C). Similar to the highly fluctuating residues, the tyrosine residues are distributed mainly on a semi-sphere around αH1-helix with Y721 and Y747 locations mostly on the y–z and x–y planes, respectively.

### 2.2. Comparative Analysis of KID^GC^, KID^D^, and KID^C^

To designate or not the KID^GC^ as a species having the qualities comparable to those of the native KID fused to the tyrosine kinase domain of KIT (KID^D^) or the cleaved KID (KID^C^), and therefore to postulate the appropriate model for the study of posttranslational processes of RTK KIT, we compared the conformational and structural proprieties of these three entities. As the MD trajectories of all KID species were generated upon strictly identical conditions that differed only in the mode of preserving its end-to-end distance, we postulated that these data might be analysed together as a unique concatenated trajectory (dataset) describing the same object—the intrinsically disordered polypeptide. Before the data analysis, all data were normalised by a fitting on the most stable structural element of KID—αH1-helix (A701-N705) taken from the KID^D^ conformation at t = 0 ns and further analysed either as all datasets, or using their selected components—a unique replica or the merged replicas for a given entity (Figure 4).

The normalised RMSDs of each analysed KID show a frequent alteration (increase/decrease) in value which are comparable between the species so that the concatenated trajectory can be viewed as a continuous 14 μs trajectory of IDP KID. The RMSD probability curves for distinct KID represent a Gaussian distribution that is partially superimposed, showing very close mean values, from 10 to 12 Å. The RMSF curves show that the minimally and maximally fluctuating residues are the same in all studied KID or at least the nearest ones. Indeed, V732 and P754 systematically display minimal RMSF values, while S717, K725, and R739 exhibit the highest.

We previously found that KID^D^ and KID^C^ display a compact globule-like shape, stabilised by a dense network of non-covalent contacts [38]. Some residues, mainly having minimal fluctuations, are more likely to be close to each other than others. Still, it is in good agreement with a broad conformational ensemble without apparent specificity between KID^D^ and KID^C^. The radius of gyration (Rg), characterising a protein shape, shows a slightly asymmetric normal distribution for KID^D^ and KID^C^ with maxima at 12.7 and 12.0 Å, respectively. In contrast, Rg of KID^GC^ shows a multimodal distribution with two equally weighted means of the bell-shaped normal distributions with maxima at 12.2 and 13.7 Å. The total number of H-bonds stabilising each KID entity (including the intra-helix contacts) is strictly identical in KID^GC^ (m.v. of 76) and KID^D^ (m.v. of 76.7). In contrast, their number is slightly reduced (3%) in KID^C^. It should be noted that the total number of van der Waals interactions is precisely the same in all the KIDs studied and that they are almost four times more numerous than hydrogen bonds.

The other metrics characterising geometry, the highly fluctuating residues—distance, pseudo-valent and pseudo-torsion (dihedral) angles—showed a substantial variance in their values (Appendix A). The values of pseudo-torsion angles defined the angle between two hyperplanes formed by the highly fluctuating residues—K725-E699-Y703-S717 (T1, synclinal), R739-E699-Y703-K725 (T2, anticlinal), and R739-E699-Y703-S717 (T3, anticlinal), are particularly interesting. The value range of the same dihedral angle is compatible for each KID entity.

### 2.3. Intrinsic Geometry of Tyrosine Residues in KID^GC^, KID^D^, and KID^C^

Focusing on the KID key residues—tyrosines—we analysed and compared the metrics related to their properties. The solvent-accessible surface area (SASA) of tyrosine residues is comparable in all studied species and between the functional phosphotyrosines—Y703, Y721, and Y730—that control KIT signalling and Y747 with the non-identified empirically function [43].

Based on our previous in silico calculations, we have assigned to Y747, located on the H4 helix, the “organising role” in the assembly of KID structure at the tertiary and quaternary level and suggested that the Y747 and αH1-helix functions are complementary and can be mutually dependent [39]. As a single Y703 is localised in the stable αH1-helix, i.e., the most conserved structural element of KID varying only in length, the other phosphotyrosines are localised on the fully transient structures, so we have supplemented for each tyrosine residue the Ramachandran plots providing an additional view on the secondary structure in each KID and their tyrosines backbone configuration (Figure 5).

The Ramachandran plot shows the statistical distribution of the combinations of the backbone dihedral angles ϕ and ψ and visualises energetically allowed and forbidden regions for the dihedral angles [44]. Typically, the permitted areas and folding of the secondary structure are residue dependent. For all non-glycine and non-proline residues of KID, the α-helices are found at m.v. of −64 ± 2 (ψ) and −41 ± 2° (φ), while the 3_10_ helices are in the upper part of the α-helices region, at −60 (ψ) and −25° (φ) [45]. For unphosphorylated tyrosines, the parallel and antiparallel β-sheets are localised in ranges of −119 ± 17 to 131 ± 16° (ψ) and −126 ± 18 to 142 ± 16° (φ), respectively. Left-handed helices are found at 60° (ψ) and 50° (φ).

The Ramachandran plot of KID tyrosine residues showed the distributions of all accessible φ and ψ values. Still, the character of these distributions is very different for tyrosines within the same KID entity and between the same tyrosine in other KID. In all KID entities, only Y703 forms a single dense maximum corresponding to the α-helical structure with a small contribution of 3_10_-helices. The unique and thick maximum observed for Y747 in KID^GC^ and KID^D^ corresponds to the area of the 3_10_-helix rather than the α-helix.

A unique but widely diffused distribution of Y721 in KID^C^ characterises its organisation into α- and 3_10_ helices. In contrast, a unique diffused distribution of Y730 in KID^D^ and KID^GC^ corresponds to an unfolded coil. Several well-resolved maxima characterise the Ramachandran plots of Y721 in the areas corresponding to a coil transiting to helix and β-strand (KID^D^), the coil transiting to helix (KID^C^), and the coil transiting to α-helix and left-handed helix (KID^GC^).

Interestingly, the β-strand area in KID^GC^ is presented by at least three distinct clusters that correspond to different types of secondary structures—parallel and antiparallel β-sheets and type II turn. The Y747 residue, considered in the literature as a rather non-functional tyrosine (non-phosphotyrosine), showed a single sharp distribution in KID^GC^ and KID^D^ corresponding to 3_10_-helix. In contrast, in KID^C^ an additional diffuse distribution around β-sheets is observed.

In general, this analysis supports the secondary structure interpretation in KID assessed in the previous works by the DSSP, which was described as a helical fold composed of the α- and 3_10_-helices [29,38,39]. Identifying helicity is in good agreement between the two methods, DSSP and Ramachandran plot. Nevertheless, the last method signalises the β-strand structures, which were not sampled by the DSSP program. Both methods confirm that the KID of RTK KIT is an archetypical intrinsically disordered entity, regardless of the context studied, either as a domain of RTK KIT or as a cleaved isolated protein, and this inherent property is manifested primarily at the secondary structure level. Each sequence segment is folded as a partially unstable (transient) structure or represents an irregular coil. Curiously, all conformational ensembles generated from different KID entities evidence that KID polypeptide tends mostly to a disordered state with a great propensity to exhibit structured transient regions.

All KID tyrosine residues, physiologically or structurally pivotal, are located in sequence regions characterised by different degrees of disorder. Only tyrosine Y703 is localised in the structurally conserved αH1-helix, varying in length, in all KID entities; Y747 is positioned on a sequence segment that is folded as regular 3_10_-helix (H4) in KID^D^ and KID^GC^ or as a partially transient structure (α-helix ↔ 3_10_-helix) in KID^C^. Two other tyrosine residues, Y721 and Y730, are located on a fully transient backbone (α-helix ↔ 3_10_-helix ↔ β-strand ↔ β-turn) in all studied KID.

The tyrosine residues are exposed to the solvent by their side chain and involved either in backbone–backbone H-bond interactions or entirely unlinked non-covalently from their environment (Figure 6).

Tyrosine Y703 showed equivalent H-bond interactions in all KID entities, either thanks to its helical folding, or similar orientation of its sidechain relative to its environment. Only the Y747 in KID^D^ is a single exception forming an H-bond by its sidechain.

### 2.4. Multiparameter Clustering of KID Conformations

KID in any studied entity is an IDP possessing transient helices interconnected with flexible loops, increasing the difficulty of regrouping structurally similar conformations based on criteria such as the RMSD. A set of 31 features (metric space) related mainly to the intrinsic polypeptide geometrical properties were selected for clustering to deliver the independence of KID-generated conformations from any referencing structure. Those algorithms require data pre-processing such as scaling and the important features selection step to improve the clustering results by discarding redundant embedded information.

First, the data were scaled between 0 and 1 for each KID entity and each metric (feature) individually. Next, the feature selection was performed by looking for high correlations/anti-correlations between feature pairs. Finally, the data dimensionality was reduced by Principal Component Analysis (PCA), keeping the first k components explaining up to 80% of the variance.

The correlation matrix constructed on these metrics revealed several correlations (Figure 7A).

Focusing on features with correlation coefficients (c.c.) ≥ 0.6 or ≤−0.6, we first observed that the S717-K725-R739 triangle area positively correlates to the distance S717-K725 (c.c. of 0.8). The rest of the considered correlation values depend solely on features involving tyrosine residues (pairwise distances and dihedral angles).

The size (volume) of the tetrahedron formed by the tyrosine residues is positively correlated with the inter-tyrosine distances Y721-Y730 and Y721-Y747 (c.c. 0.74 and 0.67, respectively). Such dependence is mainly delivered by the spatial mobility of Y721, located between the highly fluctuating residues S717 and K725, whereas Y730 and Y747 are positioned near the low fluctuating residues V732 and P754, respectively. Other correlated tyrosine features are the backbone dihedral angles. Tyrosine Y721 ψ angle is positively correlated with Y730 φ angle with a coefficient of 0.6. This indicates that the terminal Cα-atoms of fragment Y721-Y730 twist in the same direction during MD simulation. On the contrary, Y747 φ and ψ angles are anti-correlated (c.c. −0.6), suggesting a twist in the opposite direction of Y747 NH- Cα and Cα-CO planes. We identified two KID metrics (features) with high correlation/anti-correlation (≥0.8 or ≤−0.8): the distance S717-K725 and S717-K725-R739 triangle area. Keeping both does not add robust discriminating information for clustering. For further analysis, the latter was removed from the dataset.

The PCA dimensionality reduction on the remaining 30 features showed that the first two or three principal components (PCs) explain only minimal variance, 37 and 10%, respectively (Figure 7B). The most portion of variance (80%) is described by the twelve first PC, which were selected as a final dataset.

However, among more than 30 features used for clustering, five metrics representing the KID shape (the radius of gyration, Rg), the distance between the most fluctuating residues (S717 and K725), the distance between Y747 with any other functional tyrosine, and two parameters characterising the internal geometry of tyrosine (ψ angles of Y721 and Y730) clearly distinguished the clusters formed by similar conformations (Appendix A).

Thereby, C1, C2, and C5 represent le most compact clusters based on the radius of gyration, whereas C3 is the most extended. However, the tyrosine geometry and most fluctuating residues in C1, C2, and C3 show apparent differences: among the most populated cluster, C1 and C3 have a similar shape as KID ante and post-transition conformations, as observed in [39]. Despite its extended shape, C3 possesses a tighter residue-wise rectangular geometry.

Further, all generated KID conformations were classified according to their similarities using different clustering methods—DBSCAN [47], K-means algorithm [48], and hierarchical agglomerative [49]. The clustering performance was evaluated with the Silhouette score [50] and Calinski–Harabasz score [51].

A first run of the data in each algorithm on a set of hyperparameters was conducted, and their performance was calculated to find the most suitable method. The K-means method showed the best scores, followed by hierarchical agglomerative clustering and DBSCAN (Appendix A). The best agreement between scores was obtained for K-means with k = 5 and k = 6. The clustering with k = 5 gave the best Silhouette (0.35 versus 0.33), whereas k = 6 yielded the best Calinksi–Harabasz score (37,906 versus 39,536). Given the difficulty of distinguishing the optimal number of clusters based on relative performance score values, the final clustering was performed for k = 5 and k = 6.

The scores for both types of clustering performance are similar, 0.33 and 0.30 with a Silhouette and 39,536 and 37,906 with a Calinski–Harabasz score for k = 5 and k = 6, respectively. A contingency table to identify strong clusters showed that both clusterings agree well (Appendix A). The results show strong agreement for 65% of the total clustered conformations.

In particular, the cluster population strongly agrees in terms of the similarity of conformations in between two clusters found for k = 5 or 6 for C_5k = 5_/C1_k = 6_ and C3_k = 5_/C6_k = 6_ (21%), and for C1_k = 5_/C5_k = 6_ and C4_k = 5_/C2_k = 6_ (44%) (Appendix A). However, the difference between the results obtained by the two clusterings is observed in only 35% of the total conformations (C2 for k = 5 or C3 and C4 for k = 6). To avoid this ambiguity, we chose to keep the last clusters as C5 and C6, respectively. Finally, the strong population size is 22.9, 21.2, 13.8 and 7.1% for clusters C1–C4, respectively. The more ambiguous clusters, C5–C6, encompass 22.5 and 11.9% of the total clustered KID conformations.

The composition of the cluster population shows that the MD conformations of each KID object are contained in all clusters, albeit in different proportions. Cluster C1 is composed of conformations from all simulated KID (KID^D^ (1%), KID^C^ (20%) and KID^GC^ (3%); C2 comprises a mix of KID^C^ (8%) and KID^GC^ (13%); C3 and C4 are composed only of conformations issued from a lone KID entity—KID^GC^ (14%) and KID^C^ (7%), respectively. Finally, C5 and C6 are composed of a mixture of KID^D^ and KID^C^ with a prevalence for KID^D^ (17 and 11%) when the KID^C^ population represents only 6 and 1% in the respective clusters. The remaining population of conformations not regrouped into the clusters represents less than 1%. The representative conformations from the most populated clusters—C1, C2, C3, and C5—composed of the MD conformations of all KID entities are shown in Figure 7D with their gyration radii (Rg).

### 2.5. The Gibbs Free Energy (ΔG) Landscape of KID Conformations

The conformational diversity of IDP KID can also be assessed via the Gibbs free energy (ΔG) landscape as it was applied in [29,38] for KID^D^ and KID^C^. The ΔG representation provides a statistical overview of the KID conformational ensemble as a function of two reaction coordinates. It is essential to use a statistical thermodynamic treatment to analyse the data rather than assuming a two-state transition. Such treatment could be simple, but it should consider conformational entropy explicitly in terms of ensembles of microstates. Molecular simulations can test the physical significance of the choice of model used to analyse the generated data.

In our case, using rich data of the concatenated trajectory obtained by merging all trajectories of KID^GC^, KID^D^, and KID^C^ presents a rare opportunity to compile the MD conformations obtained from different KID entities simulated under similar conditions.

First, to generate the free energy landscape (FEL) of IDP KID, we used the first two principal components (PC1 and PC2) of a PCA as reaction coordinates Figure 8A. The FEL PC1 vs. PC2 shows a rugged landscape revealing KID high conformational diversity with well-defined minima indicating the multimodal distribution of both PC1 and PC2 (Figure 8B,C). The deepest well, W5, together with the adjacent low minimum W6, forms a conformational space (area 1) separated from the other (area 2) by a very high energy barrier. This splitting was created due to the bimodal profile of the PC1 component separating these two regions on a three-dimensional relief. Area 2 is complex and it displays a series of minima represented by the lowest combined well (W1–W4) and distant minima W1, W3, and W4, separated due to the multimodal distribution of the PC2 component.

Interestingly, the observed minima correspond perfectly to the multi-parameter clustering results: such clusters, C1, C3–C6 (Figure 8A), are identifiable with the deepest wells on the FEL of KID (Figure 8B,C). The wells W1 and W3 are deep but more spread out. The remaining centred extended well (W1–W4) includes conformations from C1–C4, but those clusters are well-defined with K-means.

Further, we explored the credibility of using as the reaction coordinates for FEL generation the metrics characterising the highly fluctuating residues, the pseudo-torsion angle (T), and the respective distance D between two highly fluctuating atoms of KID. As the residues S717, K725, and R739 systematically exhibit the highest RMSF values in all studied KID species (Figure 4B), and the pseudo-torsion angle characterising their relative position are correlated (Figure 4F), 2D and 3D FELs were generated using three pairs of these metrics (T versus D) (Figure 8D). The analysed residues are regularly positioned at the KID sequence, separated by 13–14 residues. In the 3D structure, these residues are also located on highly remote structural segments, but their positions are not equidistant in three orthogonal directions of KID (Appendix A). Nevertheless, we suggested that the FELs constructed using these metrics can supplement the description of the KID free energy.

All FELs are highly different, showing either multiple low minima (T1 vs. D1 and T2 vs. D2) or only the one surrounded by significantly lower wells (T3 vs. D3). The FELs with multiple minima have different profiles reflecting the anisotropic positions of the reference residues in the 3D space. High energy barriers separate the wells on each FEL.

The Gibbs free energy landscape dissected from the first principal components and geometrical metrics of the highly fluctuating residues show multiple minima, as expected for intrinsically disordered proteins.

## 3. Discussions

In this paper, we analysed KID, a crucial domain for the RTK KIT transduction process, represented by three entities: (i) a generic macrocycle (KID^GC^), (ii) a cleaved isolated polypeptide (KID^C^), and (iii) a natively fused TKD domain (KID^D^). Obtained results lead us to the postulate that these KID entities have similar structural and dynamic characteristics indicating the intrinsically disordered nature of this domain. This finding means that both polypeptides, cyclic KID^GC^ and linear KID^C^, are valid models of KID integrated into the RTK KIT and will be helpful for further computational and empirical studies of post-transduction KIT events.

Previous studies showed that KID, either as an isolated polypeptide or integrated into KIT, has a helical folding and globular shape stabilised by multiple non-covalent interactions [29,38]. The newly constructed generic cyclic KID^GC^ displayed a similar, but more compact, globular shape and was characterised by increased helical content.

The functional tyrosines Y703, Y721, and Y730 are located on a sphere of varying radii and are fully accessible to solvent. The size of the segments is different—more compact for the Y730 as for the stabiliser Y747 and very dispersed for Y721 spreading over the whole hemisphere. Moreover, in all KID species, phosphotyrosines are not stabilised by specific non-covalent bonds (with side chain contribution) that allow their full solvent availability and accessibility to protein–protein interactions and phosphate transfer.

Analysis of the tyrosine residues’ dihedral angles (Ramachandran plots) delivered additional information regarding their secondary structure interpreted by DSSP. In particular, Y703 is undoubtedly located in the same region of φ vs. ψ distribution in all studies species, corresponding to a helical folding. At the same time, the geometry of Y721 and Y730 diffuses between regions corresponding to a helical folding, coil, and β-strand with various frequencies between all species. The dihedral area ofY747 in KID^GC^ and KID^D^ is identical.

Like KID^D^ and KID^C^, KID^GC^ displays high flexibility providing its conformational diversity, as seen by the residue fluctuation profiles and clustering.

Hence, the apparent disorder of KID arises from the competition between intra-KID non-covalent interactions and high flexibility, which causes KID to dynamically alternate between sub-ensembles with different unstable fold architectures. This behaviour contrasts with the usual disorder interpretation indicative of absent tertiary interactions.

It is likely that the intrinsic disorder permits KID to bind partners via either conformational selection (fold first and then bind) [16] or induced-fit (bind first and fold while bound) [52] processes or alternating between conformational selection and induced fit [53]. Moreover, to fold upon binding as a conformational switch, KID sequences must fully encode all the structures they form in complex with diverse partners.

By grouping the generated MD conformations of all KID species according to their intrinsic characteristics, we observed the partial overlap of their conformational spaces. Therefore, whether isolated linear polypeptide, cyclised macrocycle or the domain integrated into the native KIT, KID explores similar conformations. The representative conformations of the most populated clusters show that KID has mainly a compact globular spherical shape and, less frequently, an ellipsoidal surface. These shape differences are reflected in the configuration of the tyrosine residues and the distance between the most fluctuating residues.

The Gibbs free energy landscape generated from the first principal components and geometrical metrics of the highly fluctuating residues, which form a set of KID intrinsic dynamical and geometrical features, show multiple minima as expected for intrinsically disordered protein.

We do not expect a similarity between the FEL constructed by using as reaction coordinates two principally distinct metrics—the first principal components (PC1 vs. PC2) and two geometrical measures (the inter-residue distances and pseudo-torsion angle). In computational structural biology, classical PCA reduces the big data dimensionality of extensive MD concatenated trajectories. PC1 and PC2 are the product of the eigenvectors and eigenvalues of the covariance matrix and characterise two orthogonal directions in space along which projections have the most significant variance, interpreted as the amplest atomic displacements in each MD conformation, mainly contributing to the essential dynamics [54]. Our analysis used the normalised and feature-selected dataset of intrinsic features separated from the dynamics. In contrast, the distance and pseudo-torsion (dihedral) angle describe only a subset of this dataset as the systematically measured relative geometry of the three chosen residues with the highest RMSF values.

Clustering based on over thirty features independent from any referencing structure and free-energy landscape construction on the features dataset projection on two first principal components should be best-suited to study the conformational diversity of KID of RTK KIT.

Our results demonstrate that KID^GC^ and KID^D^ display similar structural, conformational, and dynamic properties and energy-related characteristics; KID^GC^ can be used for empirical studies of KID phosphorylation and binding with its specific signalling proteins. However, since the kinase domain is a central hub of both receptor activation and communication between distant functional regions—JMR, A-loop, KID, and C-terminal, investigation of signal transduction mechanisms or the mechanisms of allosteric regulation of KIT in the native or mutated state would require full-length KIT or, at least, its full-length cytoplasmic domain.

## 4. Materials and Methods

### 4.1. 3D Modelling

The initial 3D model of KID (sequence F689–D768) was taken at 2 μs of restrained cleaved KID molecular dynamics (MD) simulation reported in [38]. Five thousand KID models completed with four glycine residues in the C-terminal were generated with Modeller 10.1 [55]. Only the GGGG motif loop was refined. The best model was chosen according to the DOPE score [56] and stereochemical quality (Procheck, Cambridge, UK) [57]. An additional five thousand models of cyclised KID^GC^ were generated from the previous loop refined KID^GC^ model using the LINK patch and assessed with the DOPE score and Procheck.

### 4.2. Molecular Dynamics Simulation

#### 4.2.1. Preparation of the Systems

For MD simulation, the model of KID^GC^ was prepared with the LEAP module of Assisted Model Building with Energy Refinement (AMBER) [58] using the ff14SB all-atom force field parameter set: (i) hydrogen atoms were added, (ii) covalent bond orders were assigned, (iii) protonation states of amino acids were assigned based on their solution for pK values at neutral pH, (iv) histidine residues were considered neutral and protonated on ε-nitrogen atoms, and (v) Na+ counter-ions were added to neutralise the protein charge.

KID^GC^ was solvated with explicit TIP3P water molecules in a periodic octahedron box with at least a 12 Å distance between the protein and the boundary of the water box. The number of atoms in the system was 20,934, 1277 and 7 for water, protein, and counter ions, respectively.

#### 4.2.2. Setup of the Systems

The setup of the systems was performed with the Simulated Annealing with NMR-Derived Energy Restraints (SANDER) module [59] of AMBER. First, the KID^GC^ system was minimised successively using the steepest descent and conjugate gradient algorithms as follows: (i) 10,000 minimisation steps with all protein atoms fixed to relax water molecules and counter ions, (ii) 10,000 minimisation steps where the protein backbone is fixed to allow protein side chains to relax, and (iii) 10,000 minimisation steps without any constraint on the system. After relaxation, the KID^GC^ system was gradually heated from 10 to 310 K at constant volume using the Berendsen thermostat [60] while restraining the solute Cα-atoms by 10 kcal/mol/Å^2^. After that, the system was equilibrated for 100 ps at constant volume (NVT) and a further 100 ps at constant pressure (NPT) maintained by the Monte Carlo method [61]. A 100 ps NPT run achieved final system equilibration to assure that the water box of the simulated system had reached the appropriate density. The electrostatic interactions were calculated using the PME (particle mesh Ewald summation) [62] with a cut-off of 10.0 Å for long-range interactions. The initial velocities were reassigned according to the Maxwell–Boltzmann distribution.

#### 4.2.3. Production of the MD Trajectories

All MD trajectories were produced using the AMBER ff14SB force field with the PMEMD module of AMBER 18 [58] (GPU-accelerated versions) running on a local hybrid server (Ubuntu, LTS 14.04, 252 GB RAM, 2× CPU Intel Xeon E5-2680 and Nvidia GTX 780ti, Canonical Ltd., London, UK) and the supercomputer JEAN ZAY at IDRIS (http://www.idris.fr/jean-zay/, accessed on 20 September 2022).

The 2 µs trajectories (replicas) were generated for KID^GC^ equilibrated system. A time step of 2 fs was used to integrate the equations of motion based on the Leap-Frog method [63]. Coordinate files were recorded every 1 ps. Neighbour searching was performed by the Verlet algorithm [64]. The Particle Mesh Ewald (PME) method, with a cut-off of 10 Å, was used to treat long-range electrostatic interactions at every time step [65]. The van der Waals interactions were modelled using a 6–12 Lennard–Jones potential. The initial velocities were reassigned according to the Maxwell–Boltzmann distribution. Coordinates were recorded every 1 ps.

#### 4.2.4. Data Analysis 

Unless otherwise stated, all recorded MD trajectories, individual and merged, were analysed with the standard routines of the CPPTRAJ 4.15.0 program [66] of AMBER 20 Suite and Python library Scikit-Learn [67]. All data analysis was performed on the MD conformations (every 10 ps) by considering all concatenated data or the production part of the simulation after least-squares fitting of the MD conformations for a region of interest, thus removing rigid-body motion from the analysis.

(1) The RMSD and RMSF values were calculated for the Cα-atoms using the initial model (at t = 0 ns) as a reference;

(2) Secondary structural propensities for all residues were calculated using the Define Secondary Structure of Proteins (DSSP) method [41]. The secondary structure types were assigned for residues based on backbone -NH and -CO atom positions. Secondary structures were assigned every 10 ps for the individual and concatenated trajectories;

(3) H-bonds between heavy atoms (N, O, and S) as potential donors or acceptors were calculated with the following geometric criteria: donor–acceptor distance cut-off was set to 3.6 Å, and the bond angle at H-atom cut-off to 120°; van der Waals contacts were considered for residues with side-chains C atoms within a 3.6 Å of each other;

(4) The mass-weighted radius of gyration (Rg) and Solvent Accessible Surface Area (SASA) were calculated for all atoms except hydrogens;

(5) The tyrosine dihedral angles (ϕ, ψ) distributions (Ramachandran plots [44]) were compared to the unphosphorylated tyrosine backbone-dependent allowed area from the Dunbrack database [46];

(6) The relative Gibbs free energy of the canonical ensemble was computed as a function of two reaction coordinates [68].
(1)$$\Delta G_{\left(R1,R2\right)}=-k_{B}T\;ln\frac{P_{\left(R1,R2\right)}}{P_{max}}$$
where *k_B_* represents the Boltzmann constant, *T* is the temperature. *P*_(*R*1, *R*2)_ denotes the probability of states along the two reaction coordinates, calculated using a *k*-nearest neighbour scheme, and *P_max_*—the maximum probability.

#### 4.2.5. Clustering

Thirty-one non-reference-dependent (intrinsic) features were calculated for the individually concatenated data of KID^C^, KID^D^ and KID^GC^ every 100 ps to create a starting dataset for clustering.

##### Pre-Processing and Features Selection

First, the data were scaled between 0 and 1 for each KID entity concatenated data and each feature.

To reduce the data dimensionality, a two-step process was applied for features selection: (i) for each pair of correlated features (coefficient of correlation should be ≥0.8 or ≤−0.8), one feature was removed from the dataset, then (ii) the first *k* PC of a PCA explaining up to 80% of the variance were kept for further analysis.

##### Clustering

First, the hyperparameters of each algorithm were tweaked to maximise the clustering performances. (i) The Density-Based Spectral Clustering (DBSCAN) [47] for a minimum sample size of 5% of the data and ε distance between 0.1 and 1 each 0.05; (ii) K-means for *k* between 2 and 15 with 1000 iterations each; (iii) Hierarchical Agglomerative clustering using Ward distance metrics and cutting the tree at *k* between 2 and 15. The best method and associated hyperparameters are then finally applied to the dataset.

The clustering performances (parameters tweaking and final clustering) were assessed with the Silhouette [50] and Calinski–Harabasz scores [51].

## Figures and Tables

**Figure 1 ijms-23-12898-f001:**
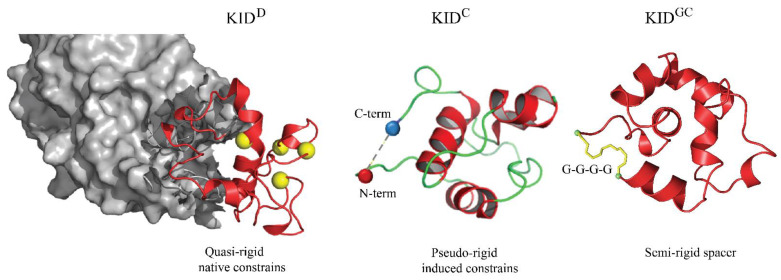
Kinase insert domain (KID) of RTK KIT as a domain fused to TKD (KID^D^), a cleaved isolated polypeptide (KID^C^) and a generic cyclic entity (KID^GC^). 3D structures are taken as randomly chosen conformations from the MD simulation of each species, with KID shown as a red cartoon, TKD as the grey surface, and the motif representing GGGG being yellow sticks. TKD is shown partially with a focus on the region adjacent to KID.

**Figure 2 ijms-23-12898-f002:**
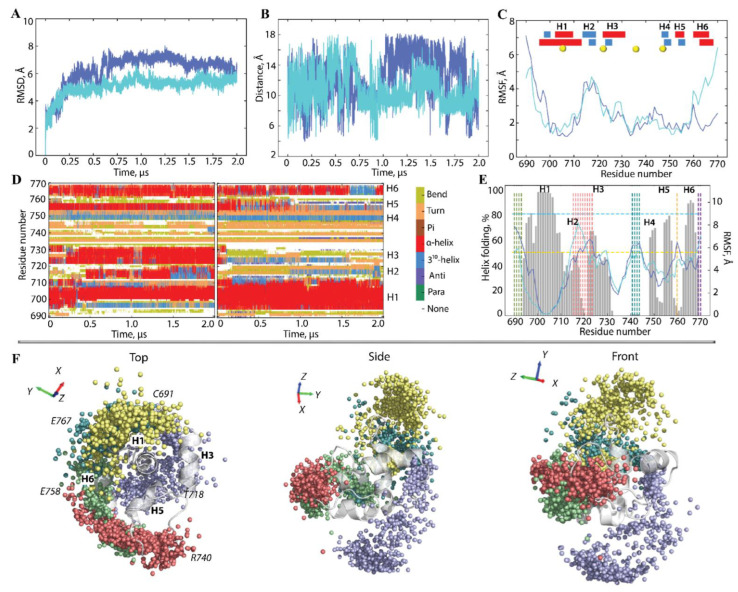
Conventional MD simulations of KID^GC^. (**A**) RMSDs, computed on the Cα atoms after fitting on initial conformation (at t = 0 ns), (**B**) distances between the Cα-atoms from F689 and D768 residues, and (**C**) RMSFs computed on the Cα atoms after fitting on initial conformation (at t = 0 ns). In (**C**), the insert shows the interpretation of the folded secondary structures, αH- (red) and 3_10_-helices (blue), labelled as H1-H6, assigned for a mean conformation of each MD trajectory. Yellow balls show the tyrosine residue position. In (**A**–**C**), MD replicas 1–2 are distinguished by colour, light and dark blue. (**D**) The secondary structure time-related evolution of each KID residue as assigned by DSSP with type-coded secondary structure bar. (**E**) Superimposition of the helical structure content (in % of the total simulation time) calculated for each residue of KID^GC^ of the concatenated trajectories and showed as grey histograms (left axis) into the RMSFs calculated after the alignment on H1-helix (A700-L706) (right axis). Coloured dashed lines trace groups of residues with higher RMSF values (>6 Å): F689-S691 in green, C714-M722 in red, R739-V742 in blue, E758 in orange, and E767-D768 in violet. (**F**) The spatial position of the highly fluctuating residues (the Cα-atoms of median residue) is distinguished by colour (C691 in yellow, T718 (1) in violet, R740 (2) in red, E758 (3) in green and E767 (in teal) projected on KID^GC^ structure after alignment on H1-helix (A701-N705). Three views—top, side and front—concerning the αH1-helix axis are shown.

**Figure 3 ijms-23-12898-f003:**
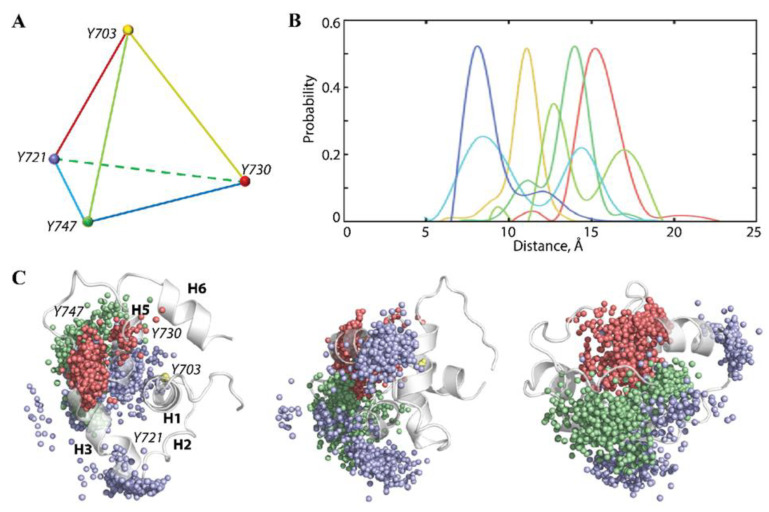
The cumulative spatial position of the tyrosine residues Y721, Y730, and Y747 relative to Y703 from the conserved αH1-helix. (**A**) The tetrahedron is defined on the Cα-atoms (vertex) of each tyrosine. (**B**) Probability of distances between each pair of tyrosine residues (tetrahedron edges). Curves are coloured as tetrahedron edges (**A**). (**C**) Position of the Cα-atoms of each tyrosine residue–Y703 (yellow), Y721 (lilac), Y730 (red), and Y747 (green), projected into the KID^GC^ 3D structure (grey cartoon) after alignment on αH1-helix (A701-N705) and shown at three orientations: top (left), side (middle), and front (right), with respect to the αH1 axe. Each point (MD frame) took 5 ns of the 14-µs MD concatenated trajectory.

**Figure 4 ijms-23-12898-f004:**
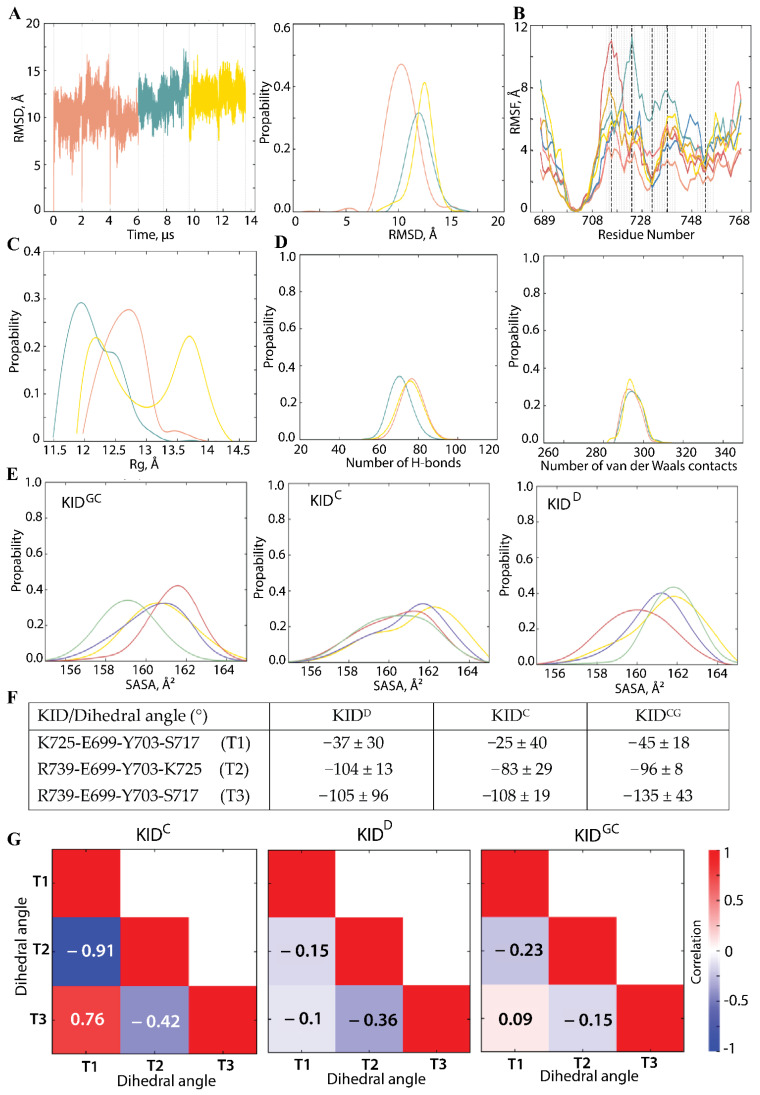
Comparative characterisation of KID using the conventional molecular dynamics (cMD) simulations of different KID entities. (**A**) RMSD was computed on the Cα-atoms (left) as well as its probability (right). (**B**) RMSF was calculated on the Cα atoms of KID^GC^ (yellow, sand), KID^D^ (red, pink, coral) and KID^C^ (teal, blue). Vertical dashed lines mark the minimally (V732 and P754) and maximally (S717, K725, and R739) fluctuating residues. (**C**) The radius of gyration (Rg) of KID entities. (**D**) Non-covalent interactions, hydrogen (H-) bonds (left) and van der Waals contacts (right), and stabilising KID entities were computed for the MD frames taken each 100 ns. Contacts with donor–acceptor (D-A) distance between heavy atoms (D and A = N, O, S) ≤ 3.6 Å, and angle at H atom (DHA) ≥ 120° were interpreted as H-bonds; distances between C-atoms ≤ 3.6 Å were attributed to van der Waals contacts. Only contacts with occurrence ≥40% were taken into consideration. (**E**) The colour denotes the solvent-accessible surface areas (SASA) of each tyrosine residue—Y703 in yellow, Y721 in violet, Y730 in red, and Y747 in green. (**F**) The dihedral (pseudo-torsion) angle is defined for the most fluctuating residues in each KID entity relative to the αH1-helix and (**G**) Correlations between the dihedral angles. (**A**–**G**) Analysis was performed on KID^GC^ (yellow), KID^D^ (red), and KID^C^ (teal) per trajectory, per entity (merged replicas), and using the concatenated data for all entities. All calculations were performed after fitting all conformations on the most stable structural element of KID—H1 (A701-N705) from the KID^D^ conformation taken at t = 0 ns.

**Figure 5 ijms-23-12898-f005:**
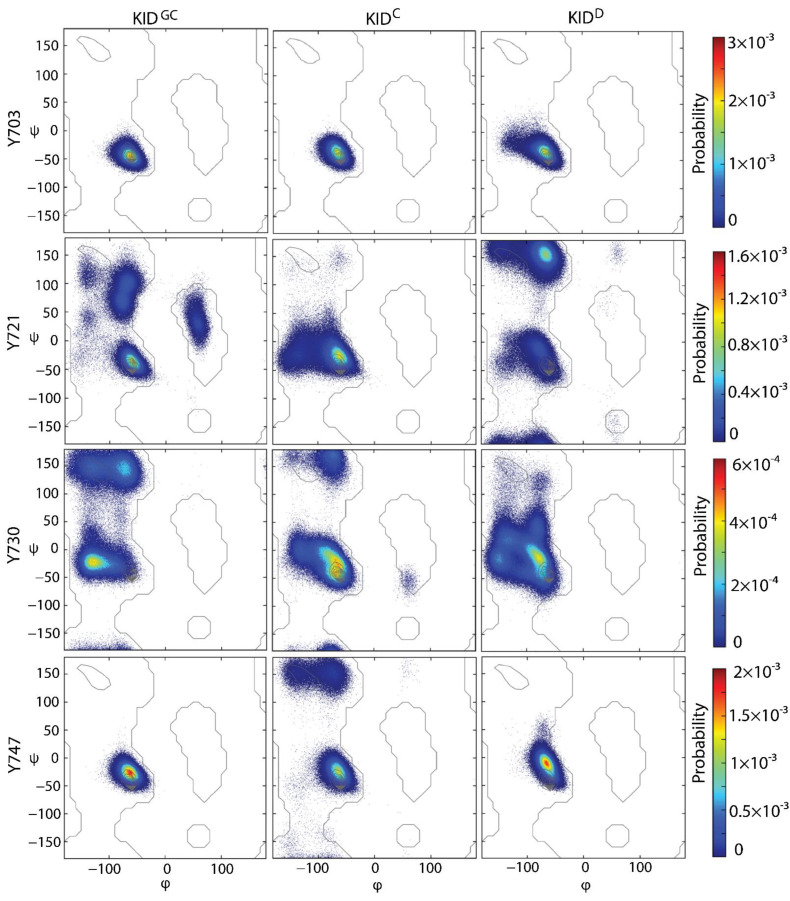
Ramachandran plots of the tyrosine residues in each KID entity. The colour scale shows the population density. Contours represent the unphosphorylated tyrosine backbone-dependent authorized regions [46]. The red colour represents high occurrence, yellow and green represent low, and blue represents the lowest occurrence.

**Figure 6 ijms-23-12898-f006:**
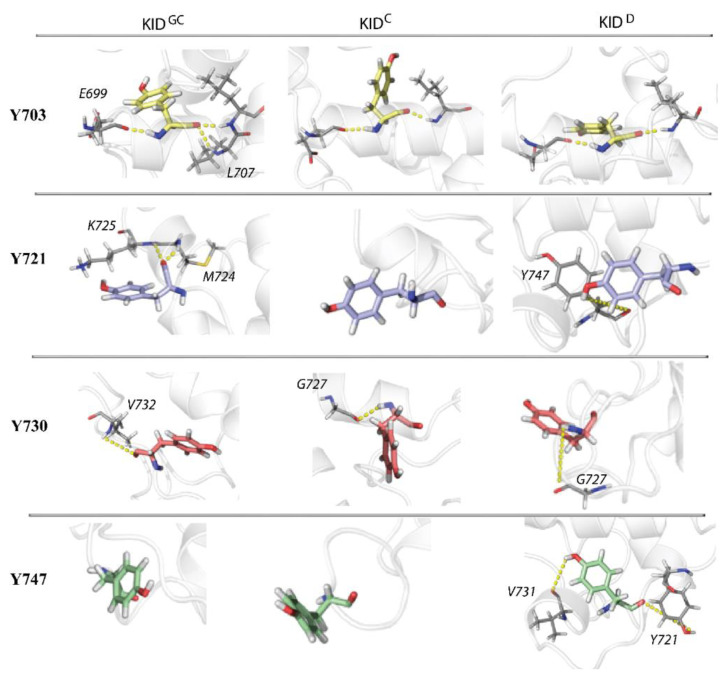
Snapshots of the KID fragments with tyrosine residues were taken at the top of each distribution (see Figure 3B). Oxygen and nitrogen atoms are in red and blue; respectively, carbon atoms are coloured differently for each tyrosine residue (Y703 in yellow, Y721 in lilac, Y730 in coral, and Y747 in green) and similarly (in grey) for other KID residues. Protein is shown as a grey cartoon. The yellow dashed lines represent the hydrogen bonds between atoms of the named residues.

**Figure 7 ijms-23-12898-f007:**
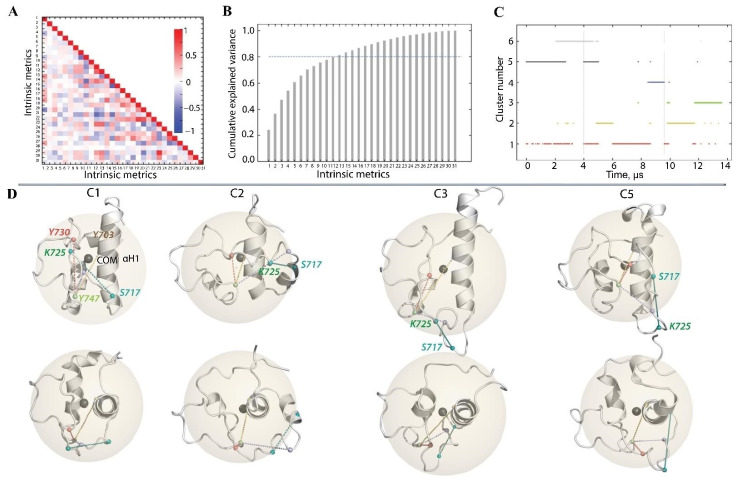
Clustering of KID conformations was performed on the concatenated 14 µs trajectory. (**A**) A correlation matrix showing correlation coefficients between 31 features. The features are (1) Rg, (2) number of hydrophobic contacts, (3) number of H-bonds, (4–7) SASA for each tyrosine residue, (8–15) backbone dihedral angles ϕ and ψ, (16–21) distances between the tyrosine residues, (22) tyrosine tetrahedron volume, (23–25) distances between the maximally fluctuating residues, (26–28) dihedral angle between the maximally fluctuating residues, (29) triangle area between the maximally fluctuating residues, (30) distance between the minimally fluctuating residues, (31) dihedral angle between the minimally fluctuating residues. (**B**) PCA analysis performed on the pre-processed data. (**C**) Clusters obtained from the concatenated 14 µs trajectory and their composition. Each colour represents different cluster. (**D**) The representative conformations from the most populated clusters are shown in three views—side and top—concerning the αH1-axis. Protein is shown as a cartoon. The sphere corresponds to the gyration radii (Rg).

**Figure 8 ijms-23-12898-f008:**
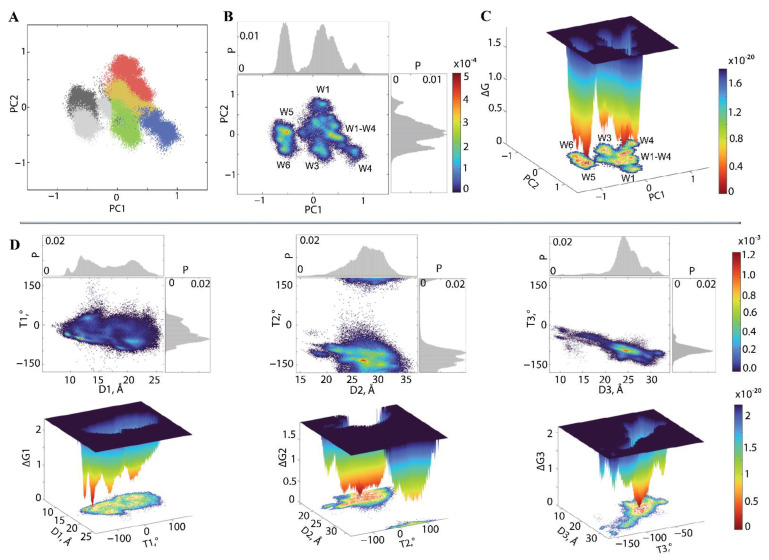
Free energy landscape (FEL) of KID in the 2- and 3-dimensional representations as a function of the reaction coordinates. (**A**) Projection of KID conformations on two principal components, PC1 and PC2. Clusters are coloured as in Figure 7C: C1 in red, C2 in yellow, C3 in green, C4 in blue, C5 in dark grey, and C6 in light grey. (**B**,**C**) FEL of KID defined on PC1 and PC2 as the reaction coordinates. (**D**) FELs of KID determined by using as the reaction coordinates the most fluctuating metrics in KID—pseudo-torsion (dihedral) angles T1 (K725-E699-Y703-S717), T2 (R739-E699-Y703-K725), and T3 (R739-E699-Y703-S717) and the respective distances D1–D3 between two highly fluctuating atoms. FELs were generated on the 14 µs concatenated trajectory composed of MD conformations of all KID entities studied—KID^C^, KID^D^, and KID^GC^. Blue represents the high energy state, green and yellow low and red represents the lowest stable state. The free energy surface was plotted using Matlab.

## Data Availability

The numerical model simulations upon which this study is based are too extensive to archive or transfer. Instead, we provide all the information needed to replicate the simulations. The model coordinates are available from L. Tchertanov at ENS Paris-Saclay.

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
