# Peer review of "Does Generic Cyclic Kinase Insert Domain of Receptor Tyrosine Kinase KIT Clone Its Native Homologue?"

_ijms, 2022, doi:10.3390/ijms232112898_

Round 1

Reviewer 1 Report

In the submitted manuscript the authors show that the different species/entities of the vital domain, KID for the Kinase Insert Domain of Receptor tyrosine kinase display comparable structures and Gibbs free energy properties. This study would enable understanding of the KID KIT RTK conformations and energy properties. The molecular dynamic simulations of the KID variants demonstrate comparable properties.  PCA show similar conformation as well. The manuscript is well written with a sufficient Introduction section and the results are well explained.  

Author Response

Reviewer 1

Comments and Suggestions for Authors

In the submitted manuscript the authors show that the different species/entities of the vital domain, KID for the Kinase Insert Domain of Receptor tyrosine kinase display comparable structures and Gibbs free energy properties. This study would enable understanding of the KID KIT RTK conformations and energy properties. The molecular dynamic simulations of the KID variants demonstrate comparable properties.  PCA show similar conformation as well. The manuscript is well written with a sufficient Introduction section and the results are well explained. 

Response: The Authors thank Reviewer 1 for very positive comments on the manuscript.

Reviewer 2 Report

In this paper, the authors studied the structural dynamics of the kinase insert domain (KID) in the tyrosine phosphorylation site of KIT, a type of RTK, by molecular dynamics simulations. The structural fluctuations of the KID loop connected with four glycine residues at the both ends were calculated and compared with those of the KID on the intact kinase domain (KIDD) and the KID fixed at each end (KIDC), obtaining various dynamics parameters.

Although the results of the calculations and analyses in this study are not questionable, the biggest problem is that the motivation for the study is ambiguous. The authors seem to claim that the dynamics of the GC structure is equivalent to that of the D and C structures. However, of course the dynamics of different molecular structures are not completely identical, and whether or not equivalence is satisfied should be evaluated according to the purpose of the calculations. The purpose of the calculation of GC structure dynamics should be explained clearly.

KID contains multiple tyrosine phosphorylation sites that are important for the signaling function of KIT. The authors' primary interest appears to be whether the accessibility of the binding proteins to these tyrosine phosphorylation sites is maintained in each structure. In reality, the active form of KIT is thought to be in a dimeric or oligomeric conformation. The question of how to assess the dynamics of KID in such a conformation by calculating the dynamics of GC and C structures should also be discussed.

A question of how the KID has evolved is also presented (line 127), but no discussion of that question is presented.

minor question

What are the colors in Fig. 8A represent?

Author Response

Reviewer 2

Response: The Authors thank Reviewer 2 for the manuscript's positive comments and critical remarks considered in the revised version. Our answers are supplied after each concern.

  1. Although the results of the calculations and analyses in this study are not questionable, the biggest problem is that the motivation for the study is ambiguous. The authors seem to claim that the dynamics of the GC structure is equivalent to that of the D and C structures. However, of course the dynamics of different molecular structures are not completely identical, and whether or not equivalence is satisfied should be evaluated according to the purpose of the calculations. The purpose of the calculation of GC structure dynamics should be explained clearly.

Response: The construction of the KID GC structure (cyclised KID) was completed after many discussions with biologists working on the signalling of KIT (and other RTKs). Two main goals were formulated: (i) to justify using a KID polypeptide having a minimal size compared to KIT, and (ii) to optimise this polypeptide so that its end-to-end distance matches that observed in native KIT. These goals are clearly explained in the manuscript (see Introduction and Discussions).

Regarding different molecular structures, we proved that all the studied objects are intrinsically disordered (ID) with similar structural and dynamic characteristics, based on the advanced statistical technics applied to big data.  This finding means that both polypeptides, cyclic KIDGC and linear KIDC, are valid models of KID integrated into the RTK KIT and will be helpful for further computational and empirical studies of KIT events after transduction.  Both objects can be used for in silico studies, while in empirical studies, the introduction of restrictions on the end-to-end distance is quite non-réalistic; therefore, for such studies, the cyclic KID is the optimal object. 

  1. KID contains multiple tyrosine phosphorylation sites that are important for the signaling function of KIT. The authors' primary interest appears to be whether the accessibility of the binding proteins to these tyrosine phosphorylation sites is maintained in each structure. In reality, the active form of KIT is thought to be in a dimeric or oligomeric conformation. The question of how to assess the dynamics of KID in such a conformation by calculating the dynamics of GC and C structures should also be discussed. A question of how the KID has evolved is also presented (line 127), but no discussion of that question is presented.

Response: Indeed, KID of KIT contains multiple tyrosine phosphorylation sites, and we estimated their accessibility in the inactive state to be phosphorylated. We are not allowed to discuss the phosphorylation effects of the active state of KIT (dimer), which makes no sense. We are currently working on analysing the active state of native KIT as a dimer and of its constitutively active mutant D816V as a monomer. A particular question is devoted to the phosphorylation of tyrosine. In these two future articles (both in preparation), we will discuss the accessibility of Ph-Tyr to protein partners (PP) and the docking of PP to the target (KID).       

  1. minor question: What are the colors in Fig. 8A represent?

Response:  Code colour for fig. 8A was added to the legend.

Finally, we revised the manuscript for English spelling.

Round 2

Reviewer 2 Report

none